# Patterns of rates of mortality in the Clinical Practice Research Datalink

**James C. F. Schmidt** [1]*, **Paul C. Lambert**[1,2], **Clare L. Gillies**[3], **Michael J. Sweeting**[1]

1 Biostatistics Research Group, Department of Health Sciences, University of Leicester, Leicester, United Kingdom, 2 Department of Medical Epidemiology and Biostatistics, Karolinska Institutet, Stockholm, Sweden, 3 Leicester Diabetes Centre, Leicester General Hospital, University of Leicester, Leicester, United Kingdom

* jcfs2@leicester.ac.uk

**Data Availability Statement:** The data used in this study were obtained from the Clinical Practice Research Datalink (CPRD), a real-world research service providing anonymised linked primary care data. The licencing agreement between the

## Abstract

The Clinical Practice Research Datalink (CPRD) is a widely used data resource, representative in demographic profile, with accurate death recordings but it is unclear if mortality rates within CPRD GOLD are similar to rates in the general population. Rates may additionally be affected by selection bias caused by the requirement that a cohort have a minimum lookback window, i.e. observation time prior to start of at-risk follow-up. Standardised Mortality Ratios (SMRs) were calculated incorporating published population reference rates from the Office for National Statistics (ONS), using Poisson regression with rates in CPRD GOLD contrasted to ONS rates, stratified by age, calendar year and sex. An overall SMR was estimated along with SMRs presented for cohorts with different lookback windows (1, 2, 5, 10 years). SMRs were stratified by calendar year, length of follow-up and age group. Mortality rates in a random sample of 1 million CPRD GOLD patients were slightly lower than the national population [SMR = 0.980 95% confidence interval (CI) (0.973, 0.987)]. Cohorts with observational lookback had SMRs below one [1 year of lookback; SMR = 0.905 (0.898, 0.912), 2 years; SMR = 0.881 (0.874, 0.888), 5 years; SMR = 0.849 (0.841, 0.857), 10 years; SMR = 0.837 (0.827, 0.847)]. Mortality rates in the first two years after patient entry into CPRD were higher than the general population, while SMRs dropped below one thereafter. Mortality rates in CPRD, using simple entry requirements, are similar to rates seen in the English population. The requirement of at least a single year of lookback results in lower mortality rates compared to national estimates.

## Introduction

Representing one of the world's largest primary care databases, the Clinical Practice Research Datalink (CPRD) contains anonymised patient level data captured at consenting general practitioner (GP) practices throughout the United Kingdom. Covering approximately 7% of the UK population, CPRD contains information on demographics, clinical results, medication usage, hospital admission, referrals, registration details and death [1]. CPRD has been shown to be representative of ethnicity, sufficiently accurate in recordings of death and comparable to other populations with regards to age and sex distribution [2–4].

University of Leicester and CPRD, and the data governance of CPRD prevent the distribution or availability of sensitive patient data to other persons. Access to the data are available from CPRD for researchers who meet the criteria for access via CPRD's Research Data Governance (RDG) Process (details at https://cprd.com/data-access).

**Funding:** This work was supported by funding from the National Institute for Health Research (NIHR) Applied Research Collaboration East Midlands (ARC EM). James Schmidt is supported by British Heart Foundation funding [grant code FS/19/11/34147]. LRWE is funded by University of Leicester, NIHR Collaboration for Leadership in Applied Health Research and Care East Midlands and Leicester Biomedical Research Centre. The Funders had no role in the study design, data collection and analysis, decision to publish or preparation of the manuscript.

**Competing interests:** I have read the journal's policy and the authors of this manuscript have the following competing interests: Michael Sweeting is a full-time employee of AstraZeneca

A common research area of Electronic Health Records (EHRs) research, including the use of CPRD, is the effect of diseases on mortality and it is therefore imperative to understand how mortality rates in a selected CPRD population compare with general population rates. The selection of cohorts on the requirement of individuals having been registered at a contributing GP practice for a specific length of time is commonplace within EHR research [5–10]. Sometimes referred to as research-quality follow-up, or *lookback window*, it is an observation period prior to the start of a subject's at-risk follow-up, ending at a date often referred to as the index date. This lookback period may be used for the clinical assessment of a comorbid condition or diagnoses, or to identify medication history. The selection effect of these delayed-entry conditions on estimated mortality rates is unknown.

In order to assess mortality rates in CPRD and the effect of the requirement for a lookback window, Standardised Mortality Ratios (SMRs) were estimated over two time scales; calendar year and follow-up period utilising CPRD data for the period 2000 to 2018.

## Materials and methods

### CPRD cohort and patient timelines

The data used comprised of CPRD GOLD patients deemed as having research acceptable data with data linkages to both the Office for National Statistics (ONS) for death registration data and secondary hospital admission data from Hospital Episode Statistics (HES). These commonly applied data linkages reduce the geographical area of CPRD to only the English data contribution. A random sample of 1 million patients was taken without replacement from research acceptable patients with data linkages to both HES and ONS, who were $\geq$18 years old and alive with CPRD follow-up after 1 January 2000. Details of the random sample and associated Stata code can be found in the S1 File. This defined the cohort entry or index date, $I(0)$, of our cohort from which mortality follow-up started (Fig 1).

A composite start date, $S$, was defined for each patient as the latest of the date of registration at their GP practice (first or current registration date) and the date the practice data was deemed to be of research quality or "up-to-standard" [11]. An end date, $E$, was defined as the earliest of the practice's last data collection date, a patient's date of transfer out of their GP

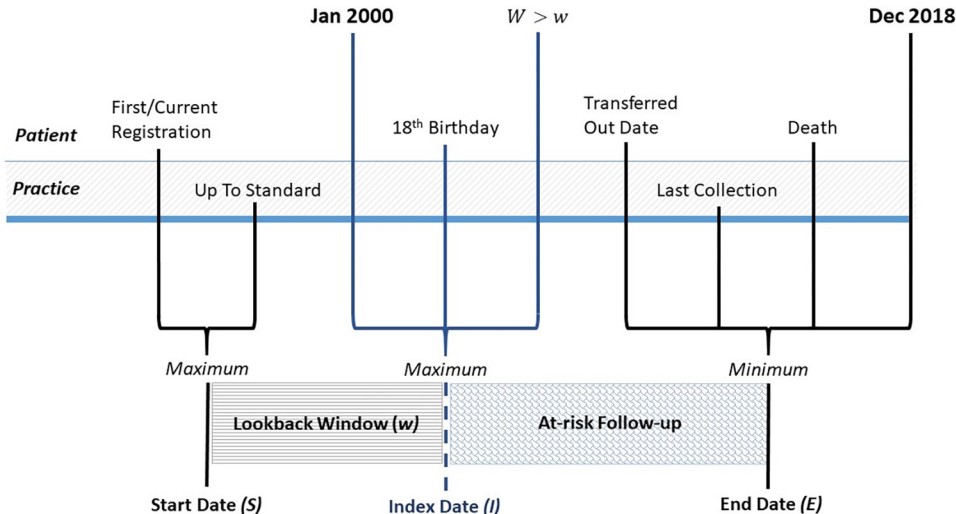

**Fig 1. Subject timelines with patient and practice level dates used to derive start date ($S$), index date ($I$) and end date ($E$).** Lookback window ($w$) and at-risk follow-up period displayed.

practice (including for death), the death date from ONS, or the administrative censoring date, 31st December 2018 (Fig 1). Four sub-cohorts were selected to have a lookback window, *W*, of *at least* 1, 2, 5 or 10 years. For each instance, a new cohort index date, *I(w)*, was defined, signifying the start of at-risk follow-up, where *W* ≥ *w*, *w* = 1, 2, 5, 10. For each new sub-cohort, those with lookback window <*w* years were omitted from the analysis. The at risk period for each individual was end date, *E*, minus the cohort index date, *I(w)*, (in years) and a crude death rate was calculated for each sub-cohort as the number of deaths divided by the total person-time at-risk, expressed per 1000 person-years. A Charlson Comorbidity Index (CCI) [12] score was calculated per patient using comorbid conditions identified in HES in the 10 years prior to cohort index date *I(w)*, baseline. The scores were classified into four groups for those with a CCI score at baseline of zero, one, two and three or more.

Reference mortality rates are derived from ONS life tables for England [13]. These published tables are based on population estimates and deaths for a three-year consecutive period. The population mortality rates used [published September 2021] covered the period 1980–1982 to 2018–2020, with the mid-year chosen to represent the data period; i.e. 2016–2018 life table captured as 2017. Life tables are stratified by age and calendar year, and published separately per gender.

## Standardised mortality ratios

The SMR is an indirect standardisation measure giving an estimate of the relative increase or decrease in mortality in a study population compared to a reference population. It is calculated as the ratio of the observed number of deaths ($D = \sum_{i=1}^{N} d_i$) within the study cohort to the expected number of deaths in the reference population (E), with $d_i = 1$ if individual *i* dies and 0 otherwise; *i* = 1,. . .,*N*. The expected number of deaths are defined as $E = \sum_{k=1}^{K} \lambda_k^* t_k$, where $\lambda_k^*$ is the mortality rate in the reference population for stratum *k*, defined by unique gender, age and calendar year combinations, and $t_k$ is the cohort's total time at-risk (measured in person-years) for that stratum. The estimation of the reference mortality rates are obtained from national actuarial life-tables published by ONS [13]. These provide precise estimates of mortality rates in the reference population, utilising mid-year population estimates and recorded mortality counts. An estimate of the overall SMR is obtained by modelling the number of observed deaths in the cohort in stratum *k*, $d_k$, such that $d_k \sim$ Poisson($E_k$), where $E_k = E[d_k] = \lambda_k t_k$ and $\lambda_k$ is the cohort mortality rate in stratum *k*. To incorporate the expected number of deaths we use Poisson regression with a log link and two offsets, $\log(t_k)$ and $\log(\lambda_k^*)$, to obtain

$$\log(E_k) = \beta_0 + \log(t_k) + \log(\lambda_k^*).$$

This gives $\theta = \exp(\beta_0) = \frac{\lambda_k}{\lambda_k^*}$ as the overall SMR, accounting for the stratum-specific mortality rates. The model can be extended to estimate stratum-specific SMRs by inclusion of explanatory variables in the Poisson regression model [14–16]. For example, we obtained estimates of calendar-year specific SMRs from data grouped by strata using the model

$$\log(E_{a,s,y}) = \beta_y + log(t_{a,s,y}) + \log(\lambda_{a,s,y}^*)$$

where $\theta_y = \exp(\beta_y) = \frac{\lambda_{a,s,y}}{\lambda_{a,s,y}^*}$ is the SMR for calendar year *y* and the subscript *a*, *s*, *y* relates to stratum combinations defined by attained age *a* (in years), sex *s*, and calendar year *y*. The individual patient data are split by age and calendar year into one-year epochs, before aggregation by unique sex, age and calendar year combination to give the total number of deaths and person-years at-risk for each stratum. The resulting aggregated data are matched with ONS published rates for the same stratum, and SMRs estimated.

## SMR by follow-up period

For the full cohort of 1 million randomly sampled CPRD GOLD patients, time-since-entry, defined as the time from index date in years (Fig 1), was included in the estimation model, providing estimates of SMRs by follow-up period. When estimating SMRs by follow-up period $f$, the data are split additionally by the third timescale, time-since-entry, defined as

$$\log(E_{a,s,y,f}) = \beta_f + \log(t_{a,s,t,f}) + \log(\lambda^*_{a,s,t})$$

The inclusion of age groups (18–59, 60–69, 70–79, 80–89, 90–99) as an interaction with follow-up period allowed for SMRs to vary by age group over follow-up period.

All analysis and modelling procedures were performed in Stata 16.

This research was approved by the Independent Scientific Advisory Committee (ISAC) for Medicines and Healthcare products Regulatory Agency Database Research (19_253RA). Generic ethical approval for observational research using the CPRD with approval from ISAC has been granted by a Health Research Authority Research Ethics Committee. Individual patient consent is not required.

## Results

Over the almost 19—year period (1st January 2000 – 31st December 2018), there were 78 729 deaths (7.9%) in the full CPRD random sample cohort (n = 1 000 000), Table 1. Each selected sub-cohort with the required lookback window $W \geq w$ [$w$ = 0,1,2,5,10], resulted in reduced cohort sizes. The sample size decreased to n = 876 048 for the sub-cohort with at least 1 year lookback, n = 771 175 for $W \geq 2$ years, n = 568 114 for $W \geq 5$ years and n = 370 780 for $W \geq 10$ years. There was some evidence of geographical variation between the sub-cohorts with the relative contribution of patients and practices from the London region decreasing for sub-cohorts with longer lookback windows. The patient pre-index CPRD history (defined as index date–start date in years) was on average 1.84 years for those with no lookback requirement, with a minimum of zero years of CPRD history, while some subjects had over 18 years of history prior to their start of at-risk follow-up. The mean pre-index CPRD history increased with increases in the lookback window requirement. Gender ratio and mean age at start date and mean age at death date remained consistent over all sub-cohorts whilst mean age at index date and end date increased with lookback reflecting an older population in the sub-cohorts. Despite this, the percentage of deaths in follow-up remained relatively consistent over sub-cohorts while follow-up decreased from over 6.5 million person-years to 2.2 million person-years from zero to ten years lookback. The mean follow-up per individual remained constant at around 6 years.

The crude death rate remained relatively stable, increasing only slightly in the ten year lookback sub-cohort. The large majority of subjects had no comorbidity at baseline across all sub-cohorts. The proportion with no comorbidity score at baseline decreased with increases in lookback, with all other comorbidity groups increasing as comorbidity burden rose due to an aging population. In those with ten years of lookback the proportion with no comorbidity reduced to 88%, compared to 91% in the sub-cohort with five years of lookback. A small increase was also seen in the mean CCI score.

Practice registration history in CPRD for patients in the full CPRD random sample (n = 1 000 000), starting when a practice is deemed to provide up-to-standard data and ending at the date of last data collection, had a mean of 16.65 (SD = 7.03) years. The longest registration was 31.6 years, while the shortest was 68 days.

Fig 2 shows the CPRD practice history, ordered from the earliest registered practices to the latest with the number of active contributing CPRD practices overlaid. The vertical red lines

**Table 1. Patient characteristics of the full cohort (W≥0) and four sub-cohorts selected by a minimum lookback window requirement.**

| | Sub-cohorts selected by a minimum lookback window | | | | |
|---|---|---|---|---|---|
| | $W \geq 0$ | $W \geq 1$ | $W \geq 2$ | $W \geq 5$ | $W \geq 10$ |
| **Subjects**[a] | 1 000 000 | 876 048 | 771 175 | 568 114 | 370 780 |
| **Pre-index CPRD History (years)**[b] | 1.84 (3.66) [0.00, 18.49] | 2.74 (3.51) [1.00, 18.49] | 3.65 (3.32) [2.00, 18.49] | 6.25 (2.62) [5.00, 18.49] | 10.41 (1.44) [10.00, 18.49] |
| **Deaths**[c] | 78 729 (7.87) | 67 540 (7.71) | 60 929 (7.90) | 46 058 (8.11) | 27 626 (7.45) |
| **Follow-up (years)**[d] | 6 539 842 (6.54) | 5 915 754 (6.75) | 5 345 168 (6.93) | 3 933 523 (6.92) | 2 186 635 (5.90) |
| **Crude Death Rate (per 1000 person-yrs)**[e] | 12.04 | 11.42 | 11.4 | 11.71 | 12.63 |
| **Charlson Comorbidity Index Score (grouped)**[c] | | | | | |
| 0 | 927 079 (92.71) | 814 348 (92.96) | 714 801 (92.69) | 519 327 (91.41) | 329 214 (88.79) |
| 1 | 42 495 (4.25) | 37 324 (4.26) | 34 143 (4.43) | 28 939 (5.09) | 23 457 (6.33) |
| 2 | 16 032 (1.6) | 13 799 (1.58) | 1 791 (1.66) | 1 563 (2.04) | 1 193 (2.75) |
| 3+ | 14 394 (1.44) | 1 577 (1.21) | 9 440 (1.22) | 8 285 (1.46) | 7 916 (2.13) |
| **Charlson Comorbidity Index Score**[f] | 0.14 (0.7) | 0.13 (0.63) | 0.13 (0.64) | 0.16 (0.7) | 0.22 (0.84) |
| **Gender**[c] | | | | | |
| *Male* | 481 866 (48.19) | 426 945 (48.74) | 379 735 (49.24) | 282 805 (49.78) | 184 942 (49.88) |
| *Female* | 518 134 (51.81) | 449 103 (51.26) | 391 440 (50.76) | 285 309 (50.22) | 185 838 (50.12) |
| **Region**[g]: | | | | | |
| *East Midlands* | 30 738 (3.07) [14] | 28 125 (3.21) [14] | 25 048 (3.25) [13] | 19 632 (3.46) [13] | 12 874 (3.47) [11] |
| *East of England* | 106 981 (10.70) [39] | 95 345 (10.88) [38] | 84 879 (11.01) [38] | 63 150 (11.12) [36] | 42 579 (11.48) [34] |
| *London* | 160 508 (16.05) [67] | 133 401 (15.23) [61] | 109 029 (14.14) [55] | 67 589 (11.90) [51] | 34 202 (9.22) [41] |
| *North East* | 18 530 (1.85) [9] | 16 915 (1.93) [9] | 15 636 (2.03) [9] | 12 992 (2.29) [9] | 10 070 (2.72) [9] |
| *North West* | 136 585 (13.66) [65] | 122 343 (13.97) [65] | 110 323 (14.31) [64] | 86 820 (15.28) [63] | 62 739 (16.92) [58] |
| *South Central* | 130 534 (13.05) [43] | 111 702 (12.75) [41] | 98 036 (12.71) [41] | 72 233 (12.71) [40] | 44 848 (12.10) [35] |
| *South East Coast* | 139 252 (13.93) [52] | 123 544 (14.10) [52] | 109 680 (14.22) [52] | 80 105 (14.10) [50] | 51 411 (13.87) [47] |
| *South West* | 124 697 (12.47) [52] | 108 380 (12.37) [51] | 95 992 (12.45) [51] | 70 967 (12.49) [49] | 44 970 (12.13) [43] |
| *West Midlands* | 116 064 (11.61) [44] | 103 340 (11.80) [44] | 92 466 (11.99) [44] | 70 677 (12.44) [44] | 49 362 (13.31) [42] |
| *Yorkshire & The Humber* | 36 111 (3.61) [17] | 32 953 (3.76) [17] | 30 086 (3.90) [17] | 23 949 (4.22) [16] | 17 725 (4.78) [16] |
| **Mean Age at**[f]: | | | | | |
| *Start Date* | 39.70 (19.86) | 39.41 (19.69) | 39.47 (19.80) | 39.30 (20.04) | 38.22 (20.23) |
| *End Date* | 48.09 (20.58) | 48.92 (20.40) | 50.06 (20.33) | 52.49 (20.19) | 54.55 (20.20) |
| *Death Date* | 78.34 (14.00) | 78.06 (13.94) | 78.03 (13.74) | 78.05 (13.33) | 78.39 (12.94) |

[Values reported are a—N, b—mean (std. dev.) [min, max], c—N (%.), d—total (mean), e–(deaths/ follow-up)x1000, f–mean (std. dev), g- mean (sdt. dev.) [unique practices]]

and shaded area demarcate the follow-up period of 01/01/2000 to 31/12/2018. Active CPRD practices providing data to CPRD rose to a peak in 2008 (n = 361) before a sharp decrease to registration levels equalling those seen in 1990 by the end of 2018.

## Lookback window and effect on SMR

The overall SMR for the 1 million CPRD random sample was 0.980 [95% confidence interval (CI) (0.973, 0.987)]. As suggested by the overall SMR, the cohort with no requirement of lookback window (w = 0) had SMRs that tended to be just below one. With increasing amounts of lookback window came reduced SMRs. The requirement of at least a single year of lookback resulted in a SMR of 0.905 (0.898–0.912). The subsequent increase in lookback revealed a trend of decreasing overall SMRs; for two years of lookback (W≥2) a SMR of 0.881 (0.874–

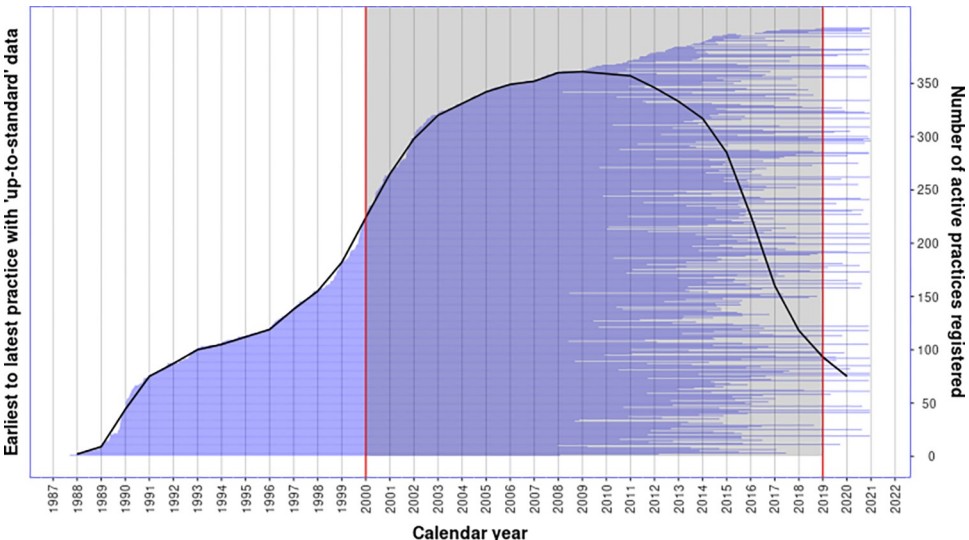

**Fig 2. CPRD practice data contribution history for GP practices associated with the 1 million random patient sample, from *up-to-standard* date to date of *last data collection*.** The shaded region shows the follow-up period with the number of active practices by calendar year overlaid (right-hand y-axis).

0.888), five years ($W \geq 5$) a SMR of 0.849 (0.841–0.857) and ten years ($W \geq 10$) a SMR of 0.837 (0.827–0.847) (S1 Table in S1 File). Across the sub-cohorts there was some evidence that the SMRs were decreasing slightly over calendar time, Fig 3.

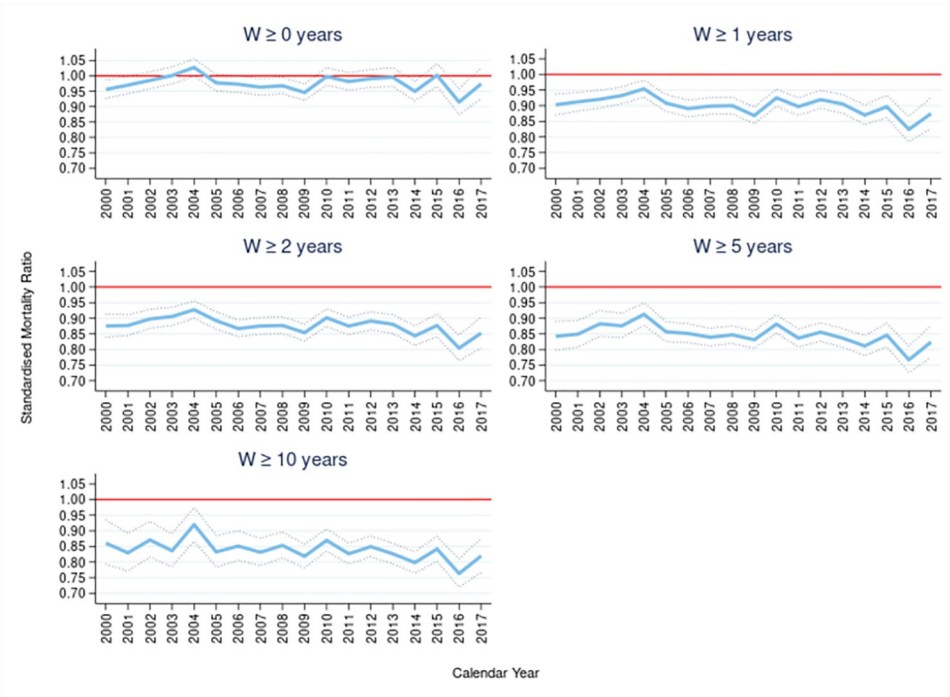

**Fig 3. Standardised mortality ratio (SMR) and 95% confidence intervals by sub-cohorts selected by a minimum lookback window $W \geq w$, over calendar year.** Reference line of SMR = 1 in red.

## Mortality by follow-up in CPRD

In the full cohort there was evidence of an initial high SMR in the first two years after entry, Fig 4 (S2 Table in S1 File). After the second year of follow-up, mortality rates reverted to below national background rates. When considered across all follow-up periods, the mortality rate in the cohort was just below the mortality rate in the general population, overall SMR = 0.980 (0.973–0.987).

## Mortality by follow-up and age group in CPRD

SMRs were estimated by follow-up and age group, Fig 5. This confirmed that the initial high SMR seen overall (Fig 4) was present in all age groups, yet the effect was lowest in the youngest age group (18–59). Older age groups had higher initial SMRs and lower SMRs in later follow-up, yet in all age groups the SMR fell below one after the third year of follow-up. This trend continued up to 19 years after study entry (index date).

## Discussion

Overall, mortality rates in the unrestricted CPRD GOLD random sample population of 1 million patients are similar to mortality rates seen in the general English population. The inclusion of a lookback window requirement of even a single year resulted in a significantly lower mortality rate in the sub-cohort once accounting for age and sex when compared with the English population. This implies that a healthier population is being selected, creating a form of selection bias. The requirement of a lookback window may inadvertently remove high-risk patients, or simply result in the selection of a more "stable" patient population. Longer registration periods with a single primary care provider may additionally result in more medically vigilant and compliant patients, all indicative of a healthier patient subgroup.

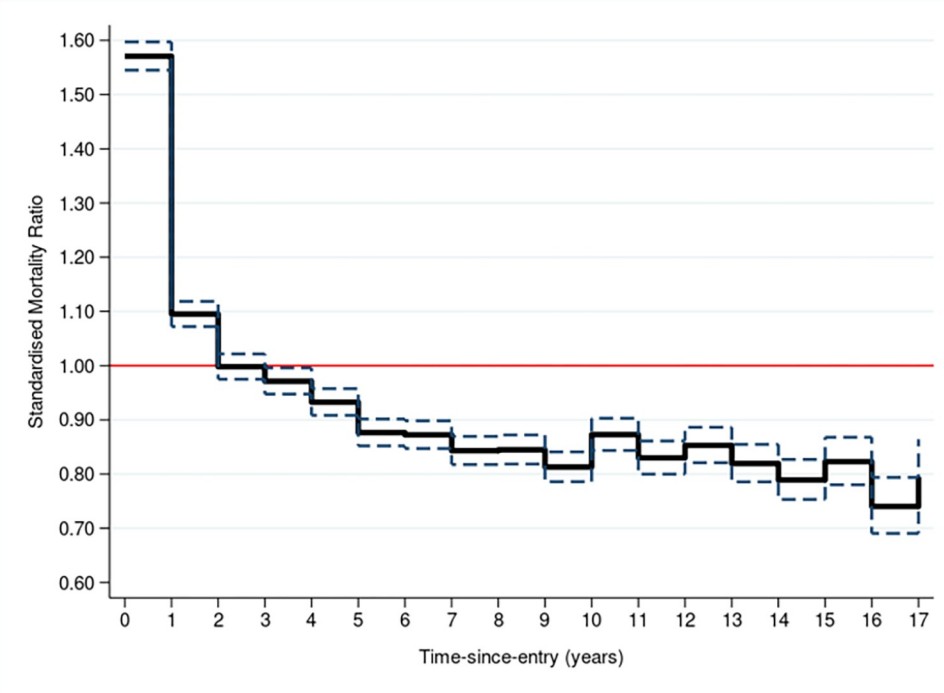

**Fig 4. Standardised mortality ratio (SMR) and 95% confidence interval by follow-up time-since-entry, in years.** Reference line of SMR = 1 in red.

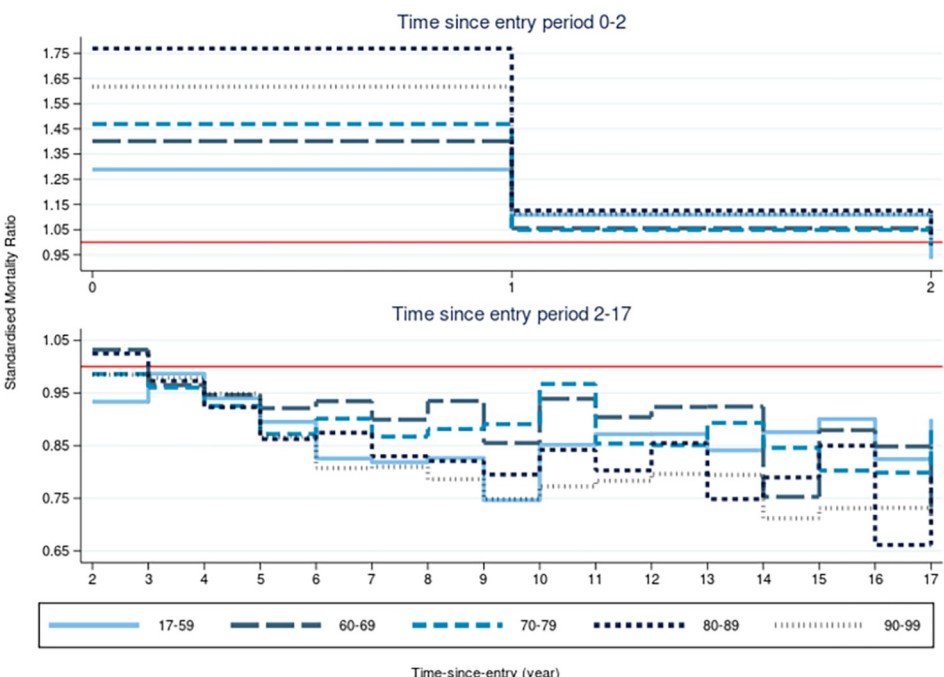

**Fig 5. Standardised mortality ratio (SMR) by age group, over follow-up period in years.** Split to show initial high mortality rate trend (5a) and lower mortality rate after year 2 (5b). Reference line of SMR = 1 in red.

The end date of a patient's follow-up, as in many EHR studies, represents a compound measure including data specific to an individual and data contributed by their registered GP practice. The end date utilised here is either the patient's date of transfer out (which can be for reasons of death), date of death, the date of last data collection from their GP practice or the administrative censoring date, whichever came earliest. As the requirement for more lookback increases, so does the proportion of patient's end dates defined by the date of last data collection from their registered GP practice. This form of censoring, though likely to be uninformative, should be examined and the impact of the selection of practices no longer contributing to CPRD considered. Similarly, the increase in lookback increases the number who reach administrative censoring, while the number of patients who transfers out of a registered GP practice decreases, emphasising the "stable" population narrative but these reasoning's may be an oversimplification of the mechanisms at play and need further investigation.

The complexity regarding the anonymity of CPRD data may be a driving factor in the high initial SMRs. Patients in CPRD represent unique lines of data. If a patient transfers out of their elected GP practice and into a new practice (for a multitude of reasons such as at their request or due to the change of residential address), this results in the creation of a "new" patient record in CPRD on registration with their new primary care provider. Therefore, it is conceivable for CPRD to contain multiple patient's records that are in fact the same individual. At current, utilising only CPRD as a data source, there is no mechanism to link these records together. It is theorised that the transfer out of patients from one GP practice and their subsequent death shortly after re-registration with a new GP practice may be accountable for a portion of the high initial SMRs seen in the first two years of follow-up.

As a hypothetical example, consider an elderly patient who transfers out of their current longstanding GP practice and moves residence into assisted care housing, registers at the

closest GP practice or a GP practice associated with the care home and then passes away 10 months after re-registration. Within the context of the data available, this would be seen as two individual records in CPRD, the first with a long CPRD record with no mortality event as the patient transferred out, and the second having a death within 10 months of registration. This hypothesis is partly supported by the finding that younger patients have lower initial SMRs than older patients do. Further investigation is needed to assess if subjects that are re-registering at a new GP practice (with previous CPRD registration history) are at a higher risk than new CPRD patients are.

A number of limitations have been identified in this research. This research was performed on a random sample of patients from CPRD and so does not represent the entirety of CPRD GOLD. Additionally, this data represented only data derived from an English population. The generalisability of these results to CPRD Aurum, other geographical areas within the United Kingdom and other large scale primary care EHRs is unknown. The lack of a full date of birth per patient, with only a birth year provided could have a marginal effect on results, while the unavailability of a linkage mechanism between de-and-re-registered patients proves vastly more problematic. The size of the sample (1 million patients) is seen as a strength though, along with the use of a robust statistical model, in the form of Poisson regression, considering changes over calendar year and follow-up, modelled on multiple time scales (age and calendar year).

## Conclusions

Regardless of the mechanism or reasoning for the selection effect or high initial mortality rates when compared to the general population, the results of reduced mortality rates with increased lookback window periods and high initial mortality rates in CPRD is significant and should be noted by all who use CPRD in the study of mortality. The use of these lookback periods is commonplace, and the implicit assumption that CPRD is representative of mortality in the general population must be carefully considered. If the requirement of lookback is consistently applied to both the study population and control group, then comparisons between groups may be valid leading to internal validity. However, when the results of a study are to be generalised to the wider population, the representativeness of the CPRD cohort should be questioned. In addition, the higher rates of mortality compared to adjusted general population rates, in the first two years of entry into CPRD, also need to be considered when addressing research questions using CPRD.

## Supporting information

**S1 File.**
(DOCX)

## Acknowledgments

The author gratefully acknowledges Leicester Real-World Evidence Unit (LRWE) for providing CPRD data. The interpretation and conclusions contained in this report/article do not necessarily reflect those of the LRWE.

This study is based in part on data from the Clinical Practice Research Datalink GOLD database obtained under licence from the UK Medicines and Healthcare products Regulatory Agency. However, the interpretation and conclusions contained in this article are those of the authors alone.

## Author Contributions

**Conceptualization:** James C. F. Schmidt, Paul C. Lambert, Clare L. Gillies, Michael J. Sweeting.

**Data curation:** James C. F. Schmidt.

**Formal analysis:** James C. F. Schmidt, Paul C. Lambert, Clare L. Gillies, Michael J. Sweeting.

**Investigation:** James C. F. Schmidt, Paul C. Lambert, Clare L. Gillies, Michael J. Sweeting.

**Methodology:** James C. F. Schmidt, Paul C. Lambert, Clare L. Gillies, Michael J. Sweeting.

**Project administration:** James C. F. Schmidt.

**Software:** James C. F. Schmidt.

**Supervision:** Paul C. Lambert, Clare L. Gillies, Michael J. Sweeting.

**Visualization:** James C. F. Schmidt.

**Writing – original draft:** James C. F. Schmidt, Paul C. Lambert, Clare L. Gillies, Michael J. Sweeting.

**Writing – review & editing:** James C. F. Schmidt, Paul C. Lambert, Clare L. Gillies, Michael J. Sweeting.

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
