## [Decision Letter · Decision Letter 0]

2 May 2022

PONE-D-22-06497Patterns of rates of mortality in the Clinical Practice Research DatalinkPLOS ONE

Dear Dr. Schmidt,

Thank you for submitting your manuscript to PLOS ONE. After careful consideration, we feel that it has merit but does not fully meet PLOS ONE’s publication criteria as it currently stands. Therefore, we invite you to submit a revised version of the manuscript that addresses the points raised during the review process.

We look forward to receiving your revised manuscript.

Kind regards,

Sreeram V. Ramagopalan

Academic Editor

PLOS ONE

Journal Requirements:

“This work was supported by funding from the National Institute for Health Research (NIHR) Applied Research Collaboration East Midlands (ARC EM).

JS is supported by British Heart Foundation funding [grant code FS/19/11/34147].”

“LRWE is funded by University of Leicester, NIHR Collaboration for Leadership in 14 Applied Health Research and Care East Midlands and Leicester Biomedical Research Centre. The interpretation and conclusions contained in this report/article do not necessarily reflect those of the LRWE.”

“This work was supported by funding from the National Institute for Health Research (NIHR) Applied Research Collaboration East Midlands (ARC EM).

JS is supported by British Heart Foundation funding [grant code FS/19/11/34147].”

Reviewers' comments:

Reviewer's Responses to Questions

**Comments to the Author**

1. Is the manuscript technically sound, and do the data support the conclusions?

Reviewer #1: Yes

2. Has the statistical analysis been performed appropriately and rigorously? 

Reviewer #1: Yes

3. Have the authors made all data underlying the findings in their manuscript fully available?

Reviewer #1: No

4. Is the manuscript presented in an intelligible fashion and written in standard English?

Reviewer #1: Yes

5. Review Comments to the Author

Reviewer #1: The authors present an interesting analysis of CPRD GOLD and contrast mortality rates with those from national morality statistics. The analysis is well described, and the results will be of interest to researchers designing studies leveraging CPRD data, including this reviewer! I only have minor comments.

1) CPRD data can come in two forms (GOLD or AURUM). My understanding is Aurum is the dominant system now. I think it’s important to acknowledge to readers that this analysis used GOLD in the abstract. Currently it is unclear until the methods.

2) Some detail on how the random sample was ascertained would be informative. It is also good practice to confirm the statistical software used for the analysis/modelling

3) The data on the subcohort characteristics is interesting. Is there some evidence of geographic variations depending on the lookback period (< in London, >in northwest, as lookback increases?). Would be interesting to understand the statistical evidence for differences in subjects in W>=0 vs the other groups. Could this be added?

4) How does a 7 year increase in mean age (between the 0 and 10 yr look back cohorts) not produce subsequent impact on mortality? That is some health selection effect. Can you explore and compare comorbidity profile (e.g. charlson comordibidity index) of the cohorts?

5) Could you overlay the line from the S1 figure on the Figure 2 via a 2nd Y axis? Tells the story concisely then?

6) Discussion covers well the questions raised by the results and your hypothetical example is helpful. One wonders whether you are able to look at cause of death to see if it is diseases of older age are driving the initial peak in SMR after 1yr of follow-up?

6. PLOS authors have the option to publish the peer review history of their article (what does this mean?). If published, this will include your full peer review and any attached files.

Reviewer #1: **Yes: **Robert Carroll

---

## [Author Response · Author response to Decision Letter 0]

21 Jun 2022

Comment 1: CPRD data can come in two forms (GOLD or AURUM). My understanding is Aurum is the dominant system now. I think it’s important to acknowledge to readers that this analysis used GOLD in the abstract. Currently it is unclear until the methods.

Response 1: Thank you for the comment. Aurum is indeed the more dominant database in terms of active patients, underpinned by the EMIS health software data capture system (EMIS Web). CPRD GOLD though has the longest coverage with the most patient follow-up. The clear distinction of the use of CPRD GOLD has been made in the abstract on page 1.

Comment 2: Some detail on how the random sample was ascertained would be informative. It is also good practice to confirm the statistical software used for the analysis/modelling

Response 2: Thank you for highlighting this lack of detail, additional detail on the method of the random sample generation has been included in the manuscript (page 3 and 6), with more expansive information and diagrams found in the supplementary material.

Comment 3: The data on the subcohort characteristics is interesting. Is there some evidence of geographic variations depending on the lookback period (< in London, >in northwest, as lookback increases?). Would be interesting to understand the statistical evidence for differences in subjects in W>=0 vs the other groups. Could this be added?

Response 3: Yes, additional data has been added to Table 1 (page 8) and results of the breakdown by region has been added to the text (page 6). 

Included in the breakdown by region is the count of patients, the percentage total and now the count of unique practices contributing data to CPRD during that lookback window. This clearly shows that a larger proportion of practices are lost in London as the restriction of longer follow-up is required. 

Comment 4: How does a 7 year increase in mean age (between the 0 and 10 yr look back cohorts) not produce subsequent impact on mortality? That is some health selection effect. Can you explore and compare comorbidity profile (e.g. charlson comordibidity index) of the cohorts?

Response 4: Thank you for highlighting this increase. We have added the crude mortality rate (per 1000 person-years) to Table 1 to highlight this issue further (page 8), along with accompanying text on the results, page 7. The mean follow-up remains constant across the lookback cohorts, at around 6 years. The total person-time at risk (in years, now included) decreases from 6.5 to 2.2 million person-years. This, coupled with a decreasing number of deaths per cohort results in a crude death rate that remains relatively stable across the cohorts. Therefore, increasing the requirement for longer pre-index date CPRD history did not result in substantially different crude death rates. However, when considering the calculation of Standardised Mortality Ratios (SMRs), rates in the sub-cohort are compared to age-, sex- and calendar year-matched rates from the general population. Here SMRs are seen to decrease with increases in lookback. As the reviewer correctly points out, with the mean age increasing in sub-cohorts we would expect mortality rates to increase. We obtain SMRs less than 1, which may be due to a healthy cohort effect with more stable patient population or more medically vigilant subgroups, as highlighted in the discussion, but these conjectures are untested.

To further describe the sub-cohorts, Charlson Comorbidity Index (CCI) scores and their categorisation into four groups (zero for no CCI score, one, two and three or more total CCI score at baseline) were assessed in the 10 years prior to the cohort index date, I(w), using Hospital Episode Statistics linked secondary care data. This showed only a slight increase in the mean CCI score for those with ten years of lookback. This is to be expected with an aging population and does not indicate a singular reason for the decrease in SMRs over sub-cohorts.

Comment 5: Could you overlay the line from the S1 figure on the Figure 2 via a 2nd Y axis? Tells the story concisely then?

Response 5: Excellent suggestion, this had now been included in Figure 2, page 9.

Comment 6: Discussion covers well the questions raised by the results and your hypothetical example is helpful. One wonders whether you are able to look at cause of death to see if it is diseases of older age are driving the initial peak in SMR after 1yr of follow-up?

Response 6: Thank you for this useful comment. The discussion highlights the impact that lookback windows have on mortality in CPRD GOLD while the exact mechanisms for this impact is unknown. We unfortunately did not receive cause of death coding with our death information from ONS. Some additional investigations, not included in this paper, investigated if certain practices were responsible for the higher initial mortality or if certain conditions could provide additional insights. It was ultimately decided that the mechanisms and reasons behind the initial high mortality would more appropriately be assigned to future work investigating this phenomenon and would not be included in this paper.

---

## [Editor Report · Decision Letter 1]

6 Jul 2022

Patterns of rates of mortality in the Clinical Practice Research Datalink

PONE-D-22-06497R1

Dear Dr. Schmidt,

We’re pleased to inform you that your manuscript has been judged scientifically suitable for publication and will be formally accepted for publication once it meets all outstanding technical requirements.

Kind regards,

Sreeram V. Ramagopalan

Academic Editor

PLOS ONE
---

## [Editor Report · Acceptance letter]

13 Jul 2022

PONE-D-22-06497R1 

Patterns of rates of mortality in the Clinical Practice Research Datalink 

Dear Dr. Schmidt:

I'm pleased to inform you that your manuscript has been deemed suitable for publication in PLOS ONE. Congratulations! Your manuscript is now with our production department. 

Kind regards, 

on behalf of

Dr. Sreeram V. Ramagopalan 

Academic Editor

PLOS ONE